# The multi scenarios applicability of GNSS differential positioning technology in the remeasurement of observatory azimuth in China

Yufei He[1,2], Xudong Zhao[1,2], Suqin Zhang[1], Qi Li[1], Fuxi Yang[3], Shaopeng He[4], Pengkun Guo[4] and Jinping Zhou[1]

[1]Institute of Geophysics, China Earthquake Administration, Beijing, 100081, China
[2]Beijing Baijiatuan Earth Sciences National Observation and Research Station, Beijing 100095, China
[3]Xinjiang Earthquake Administration, Urumqi, 830011, China
[4]Hebei Earthquake Administration, Shijiazhuang, 230071, China

*Correspondence to*: Xudong Zhao (zxd9801@163.com)

**Abstract.** The azimuth angle of geomagnetic observatory marks is crucial for ensuring the reliability of geomagnetic observation data, and its remeasurement constitutes a core task in observatory operations. The Geomagnetic Network of China comprises 46 geomagnetic observatories, most of which have been in operation for over ten years since their establishment. Thus, completing the azimuth remeasurement at these observatories has become an important mission. By comparing the precision, efficiency, and environmental adaptability of astronomical observation methods and GNSS differential positioning techniques for azimuth measurement, it is found that GNSS differential positioning technology is more suitable for the implementation of this task than traditional astronomical methods. After systematically investigating the multi scenarios in azimuth remeasurement at geomagnetic observatories based on GNSS differential positioning technology, the measurement schemes for five remeasurement scenarios are proposed, which can be applied to measurements under practical conditions such as unobstructed paths, restricted pathways, and single point deployments. Through field validations at the Hongshan, Quanzhou, and Yulin observatories, the feasibility of Scenario I (flat and clear line of sight) and Scenario II (alternative clear path available) is confirmed. Furthermore, a preliminary analysis was conducted on potential error sources in different scenarios, and a prioritized implementation sequence was established for stations that simultaneously meet the conditions of each retest scenario. This work provides a scalable technical solution for azimuth measurement in complex geomagnetic observatories environments.

## 1 Intruduction

The geomagnetic field, as a fundamental physical field of Earth, not only protects biological evolution but also serves as a key parameter in geoscientific exploration. Geomagnetic observatories, designed for long term continuous monitoring of geomagnetic variations (Jankowski and Sucksdorff, 1996), acquire seven components vector data—total intensity (F), horizontal component (H), declination (D), inclination (I), and Cartesian coordinates (X, Y and Z). These data are extensively

applied in deep resource exploration, high precision navigation, and space environment monitoring (Lu et al., 2022; Lin et al., 2023; Zhang et al., 2024a). Ensuring the precision and accuracy of absolute geomagnetic observations is of paramount importance (Zhang et al., 2024b).

It is essential to emphasize that the absolute measurement of the seven geomagnetic components at observatories is achieved through a collaborative system comprising relative recorders and absolute observation devices (St-Louis et al., 2024; Bracke et al., 2025). The relative recorders enable continuous monitoring with a sub second sampling rate, while the absolute observations calibrate baseline values periodically via manual or automated methods (currently focusing on D, I and F components twice weekly). The integration of both systems produces continuous absolute observation data with minute level

temporal resolution and accuracy better than 1 nT (Zhang et al., 2016). Consequently, the frequency and precision of absolute observations directly determine baseline reliability, which in turn impacts the quality of final data—this represents the key technical bottleneck in enhancing the accuracy of absolute geomagnetic observations within the current monitoring framework. The magnetic declination (D), defined as the angle between the true north and geomagnetic meridian, is particularly critical for practical applications such as oil drilling and navigation (Shi et al., 2008; Li et al., 2023). Currently, the measurement of

magnetic declination (D) at geomagnetic observatories is primarily conducted using fluxgate theodolites instruments (abbreviated as DI instruments), in conjunction with the azimuth angle of the observatory's reference marks. Consequently, the accuracy of the azimuth angle is fundamentally vital to the reliability of these measurements.

The azimuth mark is one of the critical core facilities at geomagnetic observatories, and the azimuth angle, defined as the angle between the true north and line connecting the center of the observation pillar to the center of the mark, is also a vital parameter.

Typically, the construction of azimuth marks requires long term stability (CEA, 2004). When feasible, the marks should ideally be engraved or built directly onto bedrock. The measurement of the azimuth angle is generally completed during the observatory's construction phase (Zhou et al., 1997; Xu et al., 2003; Yang et al., 2008; Wang et al., 2014). If the absolute observation chamber of the observatory has not yet been roofed, the azimuth angle can be directly measured at the center of the observation pillar using either astronomical observation methods (Ma, 1995; Cheng et al., 1996; Liu et al., 2020;

Khanzadyan and Mazurkevich, 2021) or GNSS differential positioning technology (Yin et al., 2008; Zhou et al., 2009; Li et al., 2015; Yu et al., 2018). The azimuth mark and angle are put into service simultaneously with the commencement of the observatory's operations.

The construction of azimuth marks requires long term stability. However, over extended periods of operation at geomagnetic observatories, these marks may undergo displacement due to natural environmental changes or human induced disturbances.

For instance, geological tectonic activities (e.g., earthquakes), crustal deformation, or subsurface fluid movements could shift observation pillars. Similarly, urban expansion—such as the construction of large scale facilities nearby, urban loading, or groundwater extraction—may lead to tectonic deformation, causing positional changes in the azimuth marks. Therefore, periodic remeasurements of the mark azimuth angles are essential in geomagnetic observatory operations to verify accuracy and promptly correct errors, thereby ensuring the reliability and precision of geomagnetic data. Azimuth angle remeasurement

is a core procedure for maintaining data quality. Consequently, the operating guidelines of China geomagnetic observatory

explicitly require the azimuth angle to be remeasured every 10 years, and if significant changes in the marks are detected, additional assessments need to be conducted immediately (CEA, 2001). The Geomagnetic Network of China consists of 46 geomagnetic observatories, most of which have been in operation for over a decade since their establishment. Thus, completing azimuth remeasurement at these observatories has become an important task.

The study first provides a concise introduction to two methods for measuring mark azimuth angles, followed by a comparative analysis. It further examines multi scenarios applicability across diverse geomagnetic observatory environments and proposes five remeasurement schemes based on GNSS differential positioning technology. Field validations at multiple observatories demonstrate the frameworks' effectiveness, offering methodological references for azimuth remeasurement protocols.

## 2 Azimuth measurement methods

Currently, geomagnetic observatories in China mainly use two methods for azimuth measurement: the traditional astronomical observation method and the modern GNSS technology. Both methods can achieve high-precision measurement results.

### 2.1 Traditional astronomical observation method

The astronomical observation method is a measurement technique that determines the astronomical longitude, latitude, and azimuth of a ground point by observing the positions of celestial bodies (such as the Sun, Polaris, or other stars) and utilizing

the relationship between their motion and the Earth's rotation. For traditional astronomical measurements, they are classified into 1 to 4 grades based on measurement accuracy, with the corresponding measurement accuracies being $0.5''$, $1.0''$, $5.0''$, and $10.0''$ respectively. In practical measurements, there are also various astronomical observation methods, including the Polaris hour angle method (suitable for grade 1-4 measurements), the solar hour angle method (for grade 4), the multi-star meridian hour angle method (for grade 3-4), the Zinger method (for grade 1-4), the solar altitude method (for grade 4), the

multi-star altitude method (grade 1-4), and so on. The celestial bodies to be observed and the observation methods are selected according to the requirements of the measurement grade.

In the northern hemisphere, especially for China, which is located in the mid-low latitudes, Polaris has a declination close to $90\,°$ and is a brightness star, making it easy to observe. Therefore, the Polaris arbitrary hour angle method is often used to measure the astronomical azimuth. However, since Polaris is not exactly at the North Pole, the precise astronomical longitude

and latitude of the observation must be known before measuring the astronomical azimuth (Liu et al., 2020). A brief introduction to the Polaris hour angle method is provided in Appendix A.

### 2.2 Modern geodetic technology

Modern geodetic technologies for azimuth are primarily implemented via the Global Navigation Satellite System (GNSS), which encompasses satellite positioning systems such as the United States' GPS, Russia's GLONASS, Europe's Galileo, and

China's BDS. Typically, GNSS employs two fundamental positioning methods: absolute positioning and relative positioning.

The absolute method provides coordinates with lower precision, so it is used for navigation purposes; in contrast, the relative method delivers coordinates with higher precision and is therefore applied in positioning scenarios. Consequently, the GNSS differential positioning method is commonly utilized in azimuth measurement. The GNSS differential positioning method employs two stationary receivers (one serving as a base station and the other as a rover) to conduct synchronized prolonged continuous observations. Through post processing error elimination and baseline resolution, this technique acquires high precision three dimensional coordinates for both measurement points (A and B). Then transforming coordinates of A and B into a common planar coordinate system (e.g., UTM or local independent coordinate system) following IGS standards. Finally, the azimuth angle is derived from the planar coordinates of reference point (A) and target point (B), using the arctangent function:

$$A = tan^{-1}\left(\frac{E_B - E_A}{N_B - N_A}\right) \tag{1}$$

where $E$ and $N$ represent easting and northing coordinate differences respectively. This method is not only fast and convenient, enabling near real time acquisition of results, but also unaffected by weather conditions, achieving weather independent operation with all weathers capability, thus exhibiting enhanced operational efficiency. Furthermore, the precision of results can be further improved by appropriately extending the observation time.

Among Modern geodetic methods, it is worth mentioning a calculation method for the azimuth remeasuremetnt based on GNSS networks. By relying on the existing GNSS observation network, this method enables azimuth measurement using only a single GNSS receiver (Sugar et al., 2012). In addition, the widespread use of electronic theodolites has also brought great convenience to azimuth observation. An automatic measurement system composed of an electronic theodolite, a GPS receiver, and a laptop with software for processing measurement data can achieve astronomical measurements that meet Grade 1-2 accuracy requirements (Zhao et al., 2003; Zhang et al., 2005; Solarić and Špoljarić, 1992; Špoljarić and Solarić, 2010).

## 2.3 Comparison and Selection

A brief comparison between the traditional astronomical observation methods and Modern geodetic technologies is presented in Table 1. Both the methods can achieve high-precision observations, yet each has its own applicable environments. Traditional astronomical observation methods have low reliance on electronic devices and are not affected by electronic signals. However, they are highly dependent on weather conditions, requiring clear skies and good visibility. In contrast, most Modern geodetic technologies rely on GNSS equipment and electronic theodolites. They are easy to operate and can work around the clock, but their performance is poor in environments with insufficient signal coverage or obstructions, and they may be affected by the multipath effect (e.g., increased errors near water surfaces, glass curtain walls, and metal reflective surfaces). According to actual observation data, the maximum GPS positioning error caused by the multipath effect can reach 3.4 cm (Fu, 2004). In high-reflection environments, this error can be as large as 15 cm (Wang, 2000). When the distance from a high-reflection surface is approximately 50 meters or more, the multipath effect can basically be neglected (He, 2010). The two methods are

irreplaceable to each other, complementing one another in terms of precision and environmental adaptability. In practical work, measurement methods can be flexibly selected according to the application scenarios.

Table 1: Comparison of Azimuth Measurement Methods

| Method | Traditional Astronomical Observation | Modern Geodetic Technology |
|---|---|---|
| Equipment | Theodolite, ephemeris, timer | GNSS receivers, electronic theodolites , data processing software |
| Limitations | Requires clear skies or visible celestial body | Requires open sky, susceptible to obstructions |
| Complexity | High (astronomy expertise needed) | Low (automated) |
| Precision | 0.5″ (Polaris based first class measurement) | 1″~2″ (ideal conditions) |
| Applications | Remote areas, military operations, heritage restoration | Engineering survey, UAV navigation, traffic planning |
| Error Sources | Atmospheric refraction, timing errors, instrument alignment | Multipath effects, ionospheric delay, satellite geometry |
| Cost | Economical | Expensive |

The Geomagnetic Network of China comprises 46 geomagnetic observatoties, most of which have been in operation for over 10 years, and their azimuth angles all require re-measurement. The mark's azimuths of the early-constructed observatories were measured using traditional astronomical methods, and usually obtained during the station construction period.  At that time, the observation rooms were not yet roofed, and theodolites could be set up directly on observation pillars to achieve unobstructed visibility of Polaris. Today, however, the rooms have been fully roofed, making it impossible to see the sky from inside. Moreover, the fixed mark pillars outdoors vary in shape (as shown in Fig.1), and are mostly unsuitable for mounting theodolites on them. Furthermore, traditional astronomical observation is highly dependent on weather conditions. Due to the diverse geographical locations of the observatories, especially those in coastal areas, where cloudy weather is frequent. It sometimes often takes several days of waiting for suitable weather to conduct astronomical method, making the measurement process time consuming and inconvenient for arranging remeasurement plans.

In traditional astronomical observation methods, the Sun can be used as the observed celestial body to measure azimuth, but it can only achieve a relatively low level of accuracy (about $10''$), as stated in the IAGA guidelines (Newitt et al., 1996). Although Solarić et al. (1988, 1990) achieved high accuracy based on a large amount of solar observation data, and pointed out that the accuracy is particularly better during sunrise and sunset periods. For high-precision observations, especially for achieving Grade 1-2 astronomical observations, and considering the current conditions of the observatories, the solar based azimuth observation method is not suitable for the re-measurement work of the 46 observatories in the network. Therefore, after considering factors such as measurement scenarios, time cost, and operational convenience, the GNSS differential positioning method was selected to complete the remeasurement work.

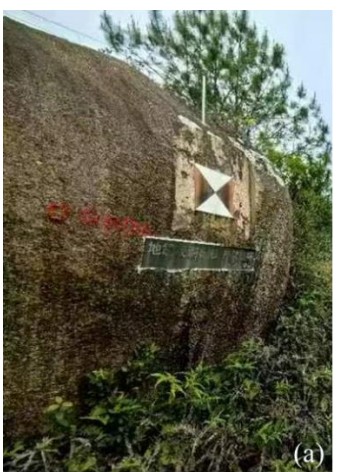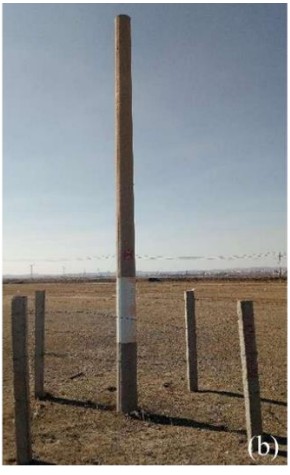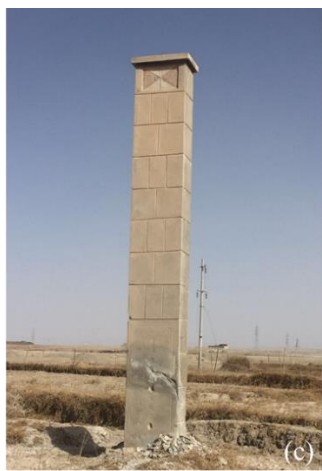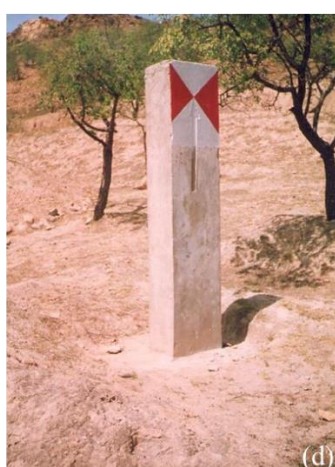

Figure 1: Different styles of azimuth marks

## 3 Multi scenarios analysis for azimuth remeasurement

In measurements based on GNSS differential positioning method, due to the varying observational conditions (such as line of sight conditions, terrain features, and surrounding vegetation coverage) across stations, require tailored measurement approaches for different scenarios. The following section discusses five common situations that may be encountered during such remeasurements.

### 3.1 Scenario I: Flat and clear line of sight

If the path between the observation pillar in the geomagnetic absolute observation room and the azimuth mark is flat and unobstructed, and free from tall vegetation coverage, two GNSS receivers can be deployed along the line of sight (LOS), with sufficient separation distance (for GNSS receivers with a horizontal positioning error of 2 mm, a minimum distance of 200 m between them is required to achieve 2″ level measurement accuracy). Additionally, the targets of both GNSS receivers are visible from the observation pillar. During this remeasurement scenario, the GNSS receivers should be installed along the unobstructed line of sight path while ensuring sufficient distance between them. A high precision theodolite (1″ level) must be used to align the azimuth mark and the two GNSS targets, guaranteeing that the center of the observation pillar, the azimuth mark, and both targets lie on a single straight line. The detailed process of installing two GNSS targets on the straight line is provided in Appendix B.

Figure 2 illustrates the schematic diagram for calculating the azimuth angle based on this method, where O represents the center of the observation pillar, A and B denote the two GNSS receiver points, M is the azimuth mark, and $a_{NM}$ and $a_{GNSS}$ indicate the azimuth angles of the azimuth mark and the GNSS receiver line, respectively (these symbols retain the same meaning in subsequent schematic diagrams and will not be reiterated in later sections). If the OM distance is sufficiently long to meet measurement requirements, points A and B can be installed between O and M, as shown in Fig. 2(a). However, if the

distance between A and B is too short to satisfy measurement criteria, point B can be relocated beyond mark M to a farther position, as depicted in Fig. 2(b).

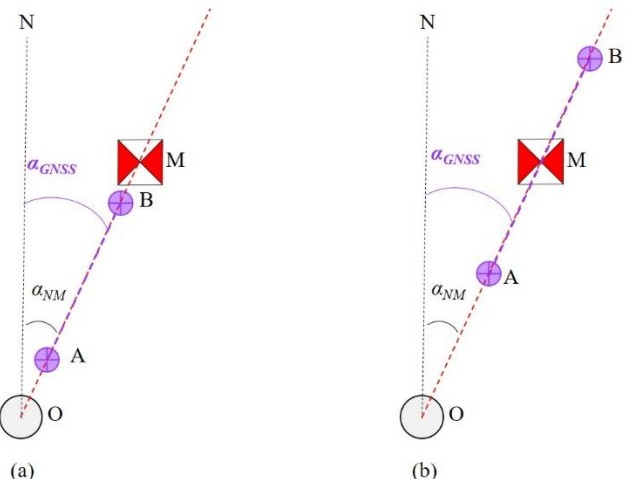

(a)           (b)

Figure 2: GNSS collinear deployment

As shown in Fig. 2, since points O, A, B, and M lie on the same straight line, the azimuth angle of the OM connection ($a_{NM}$) is identical to the azimuth angle of the line connecting the two GNSS receiver points A and B ($a_{GNSS}$). In this scenario, the azimuth angle of the GNSS receiver line is equivalent to that of the azimuth mark:

$$a_{NM} = a_{GNSS} \tag{2}$$

### 3.2 Scenario II: Alternative clear path available

When there is a clear line of sight between the observation pillar and the azimuth mark, but the path is obstructed by tall vegetation (which may interfere with GNSS satellite signal reception), or the terrain along the path is uneven (preventing visual alignment of targets from the pillar), or the path lacks sufficient distance to deploy two GNSS receivers, an alternative survey line in another direction with clear visibility and sufficient length can be identified. In such scenarios, the GNSS receivers can be deployed along this alternative survey line. By conducting GNSS measurements, the azimuth angle of this line can be determined. Subsequently, using a high precision theodolite (1″ level) to measure the angle between this line and the azimuth mark's path, the azimuth angle of the mark can ultimately be derived. Figure 3 provides a schematic diagram of this measurement method.

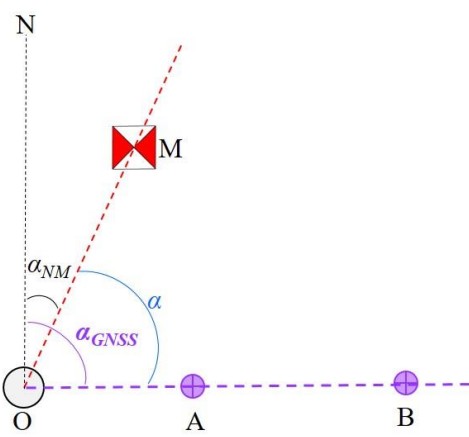

**Figure 3: Alternative path GNSS deployment**

As shown in Fig. 3, since points O, A, and B are collinear, the GNSS receivers at A and B can measure the azimuth angle of this line ($a_{GNSS}$). The angle α between the OM line and the collinear line OB can be determined through repeated measurements with a high precision theodolite (1″ level). In this scenarios , the azimuth angle of the observation mark OM ($a_{NM}$) is derived via the following formula:

$$a_{NM} = a_{GNSS} - a \tag{3}$$

### 3.3 Scenario III: Single GNSS receiver on LOS

When there is clear line of sight between the observation pillar and the azimuth mark, but insufficient distance or obstructions from tall vegetation prevents to deploy two GNSS receivers along this direction, and only one GNSS receiver is feasible. In such scenario, the following approach can be adopted. One GNSS receiver can be deployed along the path between the

observation pillar and the azimuth mark. The second GNSS receiver can be placed on another path with clear mutual visibility to the first GNSS receiver, ensuring the distance meets measurement requirements. Then the azimuth angle of the GNSS baseline can be determined using GNSS differential methods. Subsequently, a high precision theodolite, mounted on the tripod at the first GNSS receiver point, can be used to measure the angle between Point O (or Point M, depending on field setup constraints) and the GNSS target at Point B by conducting repeated measurements. The azimuth angle of the mark can be

ultimately calculated using the two angles, as illustrated in Fig. 4.

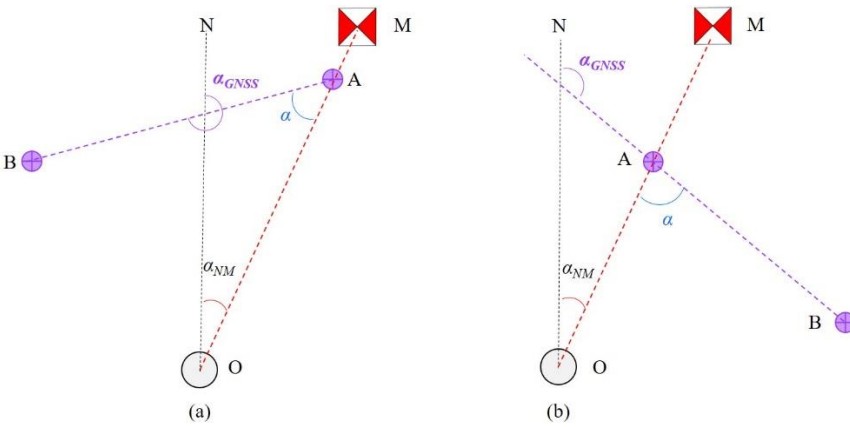

**Figure 4: Single GNSS on main path, (a) B is left of A, (b) B is right of A**

Figures 4(a) and (b) illustrate the two scenarios where Point B located to the left and right of Point A, respectively. Points O, A, and M are collinear. The azimuth angle of the baseline AB is denoted as $a_{GNSS}$. The angle between AB and AO is α. The azimuth angle of the mark ($a_{NM}$) can be calculated as follows:

$$a_{NM} = \begin{cases} a_{GNSS} - 180° - a, & (B\ is\ left\ of\ A) \\ a_{GNSS} - (180° - a), & (B\ is\ right\ of\ A) \end{cases} \tag{4}$$

### 3.4 Scenario IV: Single GNSS receiver on alternative path

If no suitable locations are available for deploying GNSS receivers along the direct path between the observation pillar and the azimuth mark, but an alternate point can be identified that maintains line of sight with both the observation pillar and another measurement point meeting distance requirements. The following workflow can be applied in this scenario. First, two GNSS receivers can be deployed at the two measurement points. The azimuth angle of the GNSS baseline connecting the two GNSS receivers can be measured using GNSS differential positioning. Subsequently, a high precision theodolite can be mounted at the GNSS receiver point which is visible to the observation pillar. Through repeated measurements, the angular offset between the second GNSS point and the center of the observation pillar is determined. The theodolite is then relocated to the observation pillar, where repeated measurements are conducted to obtain the angular between the azimuth mark and the visible GNSS point. Finally, the azimuth angle of the azimuth mark is calculated based on the angular relationships derived from these measurements. A schematic diagram of the measurement process is illustrated in Fig. 5.

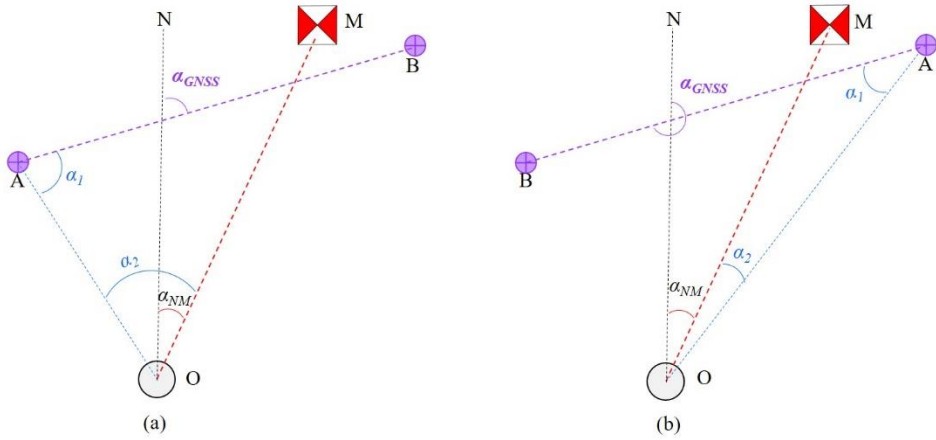


**Figure 5. Single point visibility measurement on alternative path, (a) B is right of A, (b) B is left of A**

In Fig. 5, Point A is mutually visible to both the center of the observation pillar (O) and measurement point B. Figures 5(a) and (b) depict scenarios where Point B is positioned to the right and left of Point A, respectively. Here, $a_{GNSS}$ denotes the azimuth angle measured by GNSS differential positioning, while $\alpha_1$ and $\alpha_2$ represent the angular offsets observed at Point A

and the observation pillar. Based on the angular relationships illustrated in the figures, the azimuth angle of the azimuth mark is calculated as follows:

$$a_{NM} = \begin{cases} (a_{GNSS} - 180°) - a_1 - a_2, & (B\ is\ left\ of\ A) \\ a_{GNSS} - (180° - a_1 - a_2), & (B\ is\ right\ of\ A) \end{cases} \tag{5}$$

**3.5 Scenario V: No GNSS deployment feasibility**

If none of the preceding scenarios are applicable (i.e., no GNSS compatible locations exist within the line of sight range of the

observation pillar in the observation room), but an auxiliary point can be identified that is both mutually visible to the observation pillar and aligned with a survey line that meets GNSS deployment requirements. Then the remeasurements can be completed rely on the auxiliary mutually visible point.

Most geomagnetic observatories were equipped with calibration huts containing an observation pillar. This pillar was designed to maintain mutual visibility with the observation pillar in observation room and to offer clear sightlines to its surroundings (if

Polaris could be observed from this pillar, astronomical azimuth methods could theoretically be applied, though this is not discussed here). If a specific direction near the pillar provides unobstructed visibility and meets the requirements for deploying two GNSS receivers, the pillar can serve as an auxiliary reference point. If such a pillar is not available at the observatory, a tripod can be set up at the location to act as a temporary auxiliary point.

In this scenario, a high precision theodolite is set up at the auxiliary point (either the observation pillar in the calibration hut

or a tripod mounted location). The theodolite is aligned to ensure that the two GNSS measurement points lie on the same survey line. Through repeated measurements, the angular $a_1$ between the target mark of either GNSS and the center O (marked by a fine needle secured with clay) of the observation pillar in the observation room  is determined. Subsequently, the high

precision theodolite is relocated to the observation pillar inside the observation room, where repeated measurements the angular $a_2$ between the azimuth mark and the center P of the auxiliary point. Finally, by integrating the results from GNSS differential positioning, the azimuth angle of the azimuth mark can be calculated. Figure 6 illustrates a schematic diagram of this measurement method.

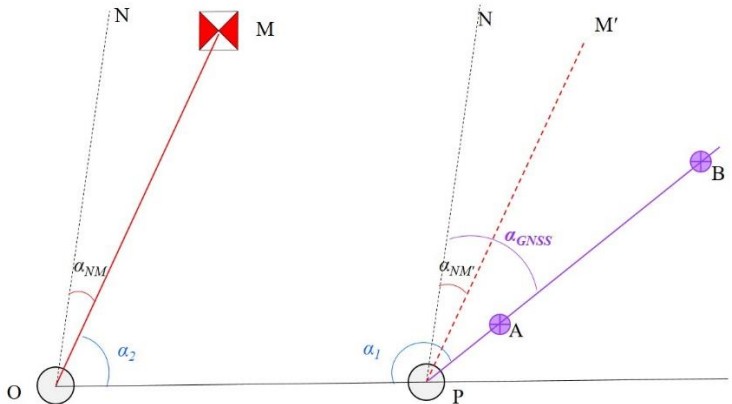

**Figure 6. Indirect measurement via auxiliary point.**

In Fig. 6, point $P$ represents the center of the auxiliary measurement point (observation pillar or tripod setup), and $PM'$ is the parallel line to $OM$. Thus, the azimuth angle $a_{NM}$ equals $a_{NM'}$. From the angular relationships shown in the figure, the azimuth angle of the mark can be derived as:

$$a_{NM} = a_{GNSS} - (a_1 + a_2 - 180°) \tag{6}$$

## 4 Multi scenarios case studies in azimuth remeasurement

The previous text analyzed five potential scenarios for azimuth remeasurement and provided corresponding measurement solutions. Based on these scenarios, we have preliminarily initiated remeasurements at stations with long standing azimuth marks and obtained initial results. Currently, azimuth remeasurements have been completed at three stations: Hongshan Geomagnetic Observatory, Quanzhou Geomagnetic Observatory, and Yulin Geomagnetic Observatory.

Hongshan Geomagnetic Observatory, located in the North China Plain, started its previous azimuth mark in 2003 (22 years ago); the path between its observation pillar and azimuth mark features flat terrain with a completely unobstructed line of sight, satisfying Scenario I conditions, hence Scenario I methodology was adopted. Quanzhou Geomagnetic Observatory in southeastern China's hilly terrain has its azimuth mark engraved on bedrock, dating back to 2007 (18 years ago); although the path is complex, two measurement point meeting Scenario I requirements can be identified, so Scenario I was applied. Yulin Geomagnetic Observatory, located in the suburban area of Yulin, northwestern China, had its original primary azimuth mark destroyed. However, before its destruction, the azimuth angle was transferred to the rooftop of a distant building and has been in use since 2009 (16 years ago). Due to multiple buildings and tall trees obstructing the path, measurements using Scenario I

were unfeasible. Fortunately, an alternative survey line meeting Scenario II requirements was identified from the opposite side of the observation room, enabling azimuth remeasurement via Scenario II. Additionally, Scenario IV methodology was also attempted in Yulin observatory. Detailed results for all three observatories are presented in Table 2.

**Table 2. Comparison of remeasured and original azimuth angles**

| Observatory | Scenario | Pillar No. | Remeasured (GNSS) | | Original (Astronomy) | Difference |
|---|---|---|---|---|---|---|
| | | | Azimuth | Standard deviation | Azimuth | |
| Hongshan | I | 1# | 1° 31′ 55.8″ | ±2.1″ | 1° 31′ 52.2″ | 3.6″ |
| | | 2# | 2° 22′ 10.8″ | ±1.6″ | 2° 22′ 9.6″ | 1.2″ |
| Quanzhou | I | 1# | 179° 46′ 37.9″ | ±0.7″ | 179° 46′ 28.6″ | 9.3″ |
| | | 5# | 178° 30′ 34.4″ | ±0.8″ | 178° 30′ 25.6″ | 8.8″ |
| Yulin | II | 1# | 354° 49′ 40.2″ | ±0.7″ | 354° 49′ 31.2″ | 9.0″ |
| | IV | 1# | 354° 48′ 49.3″ | ±0.7″ | | 41.9″ |


In the azimuth remeasurement work at these three geomagnetic observatories, we uniformly used the GNSS equipment with the rapid static horizontal accuracy 2.5mm+0.5ppm RMS, employed the Zeiss Theo 010B theodolite with an angular measurement accuracy of $1^{cc}$ (0.324″), war used for collinearity alignment and angle measurement. Additionally, three sets of data were collected for each GNSS survey line (in the WGS84 coordinate system), and the mean value and standard deviation are presented in Table 2. Most of the differences between the re-measured azimuths and the original astronomical azimuths are within the range of 0-9″. However, in the Scenario IV test at Yulin observatory, a significant deviation (41.9″) was observed between the measured and original results. Three primary factors may contribute to this result. The first one is the insufficient GNSS baseline length. Despite the RMSE of three measurement sets remained stable at 0.7″, the short distance between two GNSS points (about 60 m) and the maximum horizontal error (about 2.5 mm) may introduce the potential angular error of up to 8.6″. The second one is the theodolite setup error. In this scenario, the theodolite needs to be installed on the tripod at point A, as shown in Fig. 5(b). During angle measurement process, a centering offset was detected on the tripod. After re-centering, a change of 10.8″ occurred. This underscores the necessity of adequate baseline distances and highlights the need for further analysis of tripod measurement error characteristics. The last one is that measurement errors will be introduced when measuring the angles between the azimuth mark and the GNSS target in Scenario II and IV using theodolite. In this work, each angle was measured 6 times. The standard deviation of angle $a$ in Fig. 3 of Scenario II is 2.5″, and standard deviation of angle $a_1$ and $a_2$ in Fig. 5b of Scenario IV are 3.3″ and 1.8″, respectively. It can also be noted that the standard deviation of the angle ($a$ and $a_2$) measured on the observation pillar is smaller than that ($a_1$) on the tripod. Therefore, the stability of observation points is also important. Compared with the difference in Table 2, the measurement errors cannot be ignored in

the analysis. In order to reduce the error introduced by the measured angle, it is necessary to increase the number of repeated
measurements.

It should be noted that the azimuth obtained by the astronomical observation method is referenced the plumb line, while the azimuth obtained by the geodetic method (derive from the GNSS method) is based on the ellipsoidal normal. So the vertical deflection introduces systematic error to the azimuths (Vittuari et al., 2016). For example, Šugar et al. (2012) compared azimuths obtained from astronomical methods and GNSS measurements, finding an average difference of 0.8″, and the
standard deviation of the astronomical azimuth was ±2.6″. Based on 14 astronomical observations and comparative experiments with GNSS, Wang et al. (2001) reported that the differences between the two methods mostly ranged between 0–3″, with a maximum difference of up to 4.5″.

Accounting for the influence of vertical deflection, it is necessary to apply the Laplace azimuth equation (7) to convert the astronomical azimuth to the ellipsoidal normal based azimuth (Torge, 2001),

$$a - A = -\eta \tan \varphi + \frac{\eta \cos \alpha - \xi \sin \alpha}{\tan \zeta} \tag{7}$$

where A is the azimuth determined by astronomic observation and referred to the plumb line, $a$ is the azimuth referred to the ellipsoid normal, $\zeta$ is the ellipsoidal zenith-distance to the reference mark, $\eta$ and $\xi$ are the components of the deflection of the vertical: $\xi = \Phi - \varphi$ and $\eta = (\Lambda - \lambda) \cos \varphi$. They can be calculated by the astronomic coordinates $(\Phi, \Lambda)$ and the ellipsoidal coordinates $(\varphi, \lambda)$. Due to the lack of astronomical latitude $\Phi$ and longitude $\Lambda$ values for the measured points, it is temporarily
impossible to conduct this calculation. However, the global gravity field model provides a way to calculate the vertical deflection. Based on the gravity field model EGM2008 (https://icgem.gfz-potsdam.de/), the vertical deflections at the measurement point were calculated in the WGS84 coordinate system, including the north-south component $\xi$ and the east-west component $\eta$. After providing the zenith distance $\zeta$ of the azimuth marker, the geodetic azimuth of the azimuth mark can be finally calculated according to formula (7), and then the difference with the GNSS results can be computed, as shown in
Table 3. It can be seen that the difference between the converted azimuth and the re measured azimuth has decreased, with the difference generally within 5″. Therefore, when considering systematic deviations between astronomical and GNSS methods, positioning errors, sighting alignment errors, and other factors, a discrepancy of 0–5″ is within expected limits. In addition, assuming that the azimuth mark has shifted, for an azimuth mark located 200m from the observation pillar center, it would need to be displaced 5 mm in the direction perpendicular to the line connecting the two to achieve a 5″ discrepancy (the
displacement occurring in the direction of the connection between the two cannot be detected). Furthermore, in terms of the accuracy requirement for azimuth in absolute geomagnetic observation (0.1′), a variation of 0–5″ over several decades is considered acceptable. Thus, it can be concluded that no significant displacement has occurred at the observation marks. The previous measurement results can also serve as reference for results evaluation.

**Table 3. The converted azimuth angle after vertical deflection correction and its difference from CNSS results**

| Observatory | Scenario | Pillar No. | Vertical deflection | | Zenith Distance | Corrected Angle | Converted Azimuth | Difference |
|---|---|---|---|---|---|---|---|---|
| | | | $\eta$ (″) | $\xi$ (″) | $\zeta$ (°) | $\Delta A$ (″) | $a$ | |
| Hongshan | I | 1# | -1.57120 | -2.45921 | 89.98065 | 1.20077 | 1°31′53.4″ | 2.4″ |
| | | 2# | | | 89.97359 | 1.20060 | 2°22′10.8″ | 0.0″ |
| Quanzhou | I | 1# | -8.49674 | 3.83258 | 86.44249 | 4.48940 | 179°46′33.1″ | 4.8″ |
| | | 5# | | | 86.24792 | 4.51258 | 178°30′30.1″ | 4.3″ |
| Yulin | II | 1# | -8.86352 | -2.51050 | 89.96663 | 7.01978 | 354°49′38.2″ | 2.0″ |
| | IV | 1# | | | | | | 48.9″ |

## 5 Conclusions

This paper proposes five applicable remeasurement scenarios based on GNSS differential positioning, comprehensively covering all remeasurement challenges. Comparative analysis reveals: Scenario I requires only GNSS receiver alignment
(ensuring collinear points), simplifying operations without angular measurements or conversions, thus introducing no additional errors beyond four points alignment inaccuracies. Scenarios II and III necessitate measuring one angle (beyond alignment), introducing single error sources. However, Scenario II performs this measurement on more stable observation pillars, while Scenario III uses tripods, making Scenario II superior. Scenarios IV and V require measuring two angles (introducing dual errors), with one typically tripod based. If Scenario V employs a stable observation pillar as an auxiliary
point, it outperforms Scenario IV. Consequently, based on the principle of 'one more measurement, one more error introduced', in order to minimize the introduction of errors, the priority order of measurement schemes for remeasurement is as follows: Scenario I (optimal) > Scenario II > Scenario III > Scenario V (with fixed pillar) > Scenario IV.

Field measurements at three observatories using Scenarios I and II confirm the feasibility of these remeasurement methods. Remeasurements show minimal azimuthal changes at all three observatories. Quanzhou's mark is engraved on bedrock;
Hongshan's mark is mounted on a pillar with a stable foundation on level ground; Yulin's original mark was destroyed, and its mark relocated to a distant rooftop (relatively stable but weather vulnerable and not recommended for permanent marks). These cases highlight that mark stability of azimuth marks is crucial.

A slight change of azimuth mark was detected at these three observatories. Due to the small changes, which is basically within the measurement error range, but still meet the requirements of the magnetic declination accuracy of the geomagnetic
observatories, it can be concluded that the azimuth markers at all three stations remained unchanged. However, in future measurements, particularly for angle measurements in scenarios other than Scenario I, increasing the number of measurement repetitions is highly recommended to improve observation accuracy. In any case, these findings enhance the data accuracy assurance for observatory operations and verify the importance of this work.

## Appendix A: Polaris hour angle method

Figure A1 illustrates the angular relationships among the azimuth mark, celestial body, and true north in the horizontal coordinate system centered at observation point $O$. In this system: $Z$ represents the zenith, $M$ denotes the ground azimuth mark, $S'$ is the projection of celestial body $S$ onto the horizontal plane, $P$ stands the north pole, $N$ is the intersection point of the great circle passing through the zenith and the north pole with the horizon. $A$ indicates the celestial body's azimuth angle, and $\theta$ signifies the horizontal angle between the celestial body and the mark. To determine the azimuth angle $(a_{NM})$ of OM,

the following two steps are required: (a) measure the horizontal angle $\theta$ between the celestial body and the mark; (b) record the observation time and calculate the celestial body's azimuth angle $A$. Finally, the azimuth angle $a_{NM}$ can be derived as follows:

$$a_{NM} = A + \theta \tag{A1}$$

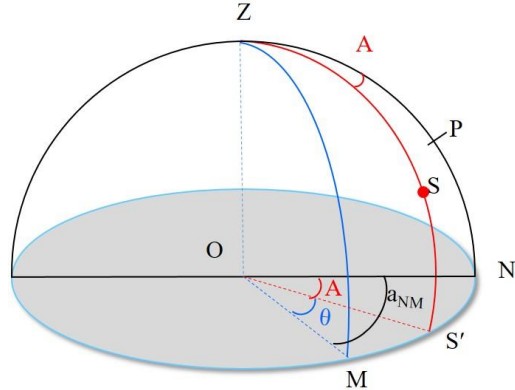

**Figure A1: Spherical Diagram of Astronomical Azimuth**

Figure A2 shows the spherical triangle $\triangle ZPS$ used to calculate the azimuth of Polaris. Based on the altitude angle h of Polaris in the horizontal coordinate system, the latitude $\varphi$ of the observation point, the declination $\delta$ of Polaris, and its hour angle $t$, the formula for calculating the celestial azimuth A (with the positive direction being eastward from north) can be derived using spherical trigonometry:

$$A = \tan^{-1}\left(\frac{-\sin t}{\cos \varphi \tan \delta - \sin \varphi \cos t}\right) \tag{A2}$$

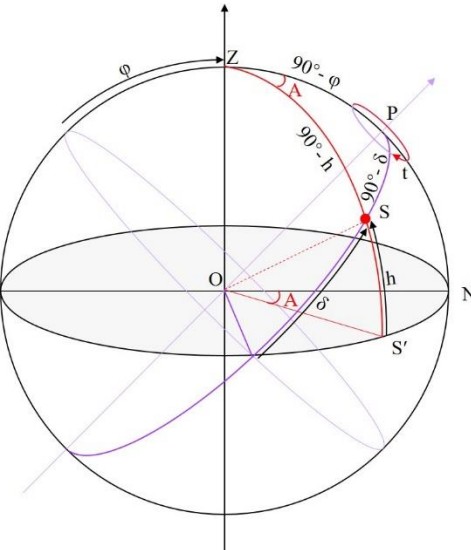

**Figure A2: Spherical triangle for calculating the celestial azimuth**

The celestial azimuth A can also be accurately derived using software (such as NOVAS developed by https://aa.usno.navy.mil/software/novas_info), where the irregular fluctuations of Earth rotation is corrected by reading the Earth orientation parameters. After obtaining the true northern azimuth of the Polaris through formula (A2) and measuring its horizontal angle $\theta$ with the ground mark, the azimuth of the mark can be determined (GB/T 17943-2000).

## Appendix B: Aligning two GNSS targets method

To achieve collinearity among the four elements, the following operational procedures must be strictly followed: First, set up the theodolite on the observation pillar inside the observation room, and precisely adjust it to ensure the instrument's center aligns perfectly with the pillar's center. Then, perform meticulous theodolite leveling. After leveling, aim at the azimuth mark through the theodolite's telescope, and once the crosshairs coincide exactly with the azimuth mark's center, lock the theodolite's horizontal circle at this position (while keeping the vertical circle freely rotatable). Next, mount the leveling base and GNSS target on a tripod, employing a two-person coordination method—one moving the tripod and the other observing through the theodolite's telescope. Position the tripod within the telescope's field of view, then gradually adjust its placement until the GNSS target infinitely approaches the line defined by the crosshairs and the azimuth mark. Secure the tripod afterward. At this point, adjust the leveling base. After leveling, the GNSS target may deviate from the line, requiring another fine adjustment to realign the mark, followed by re-leveling. This cycle must be repeated multiple times until the base is perfectly level and the GNSS target precisely lies on the line. Only after fulfilling these conditions can the GNSS receiver be mounted on this tripod system. The same procedure is applied to the second tripod. Ultimately, ensure the four elements are rigorously collinear. As shown in Fig.B1, images illustrate the alignment process from the observation pillar to the GNSS targets and the

azimuth mark. Figure B1(a) demonstrates azimuth mark alignment: After precisely centering the azimuth mark using the telescope crosshairs, the horizontal circle should be locked in place. Figure B1 (b) shows initial positioning: The tripod with the GNSS target should be aligned with the sightline formed by the telescope crosshairs and the azimuth mark. Figure B1 (c) depicts the first aligned GNSS receiver, while Fig. B1 (d) presents the second aligned GNSS receiver and azimuth mark.

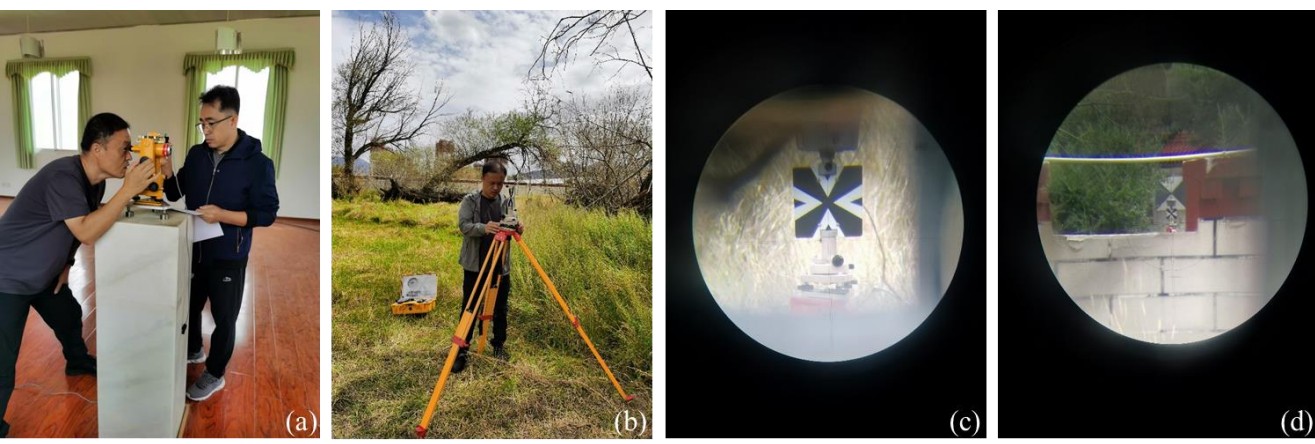


**Figure B1: Theodolite alignment to the GNSS targets and the azimuth mark.**

### Data availability

All raw data can be provided by the corresponding authors upon request.

### Author contribution

YH and XZ initiated the study. SZ and QL designed the analysis methods. FY, SH and PG carried them out. JZ provided photos of the azimuth marks. YH prepared the manuscript with contributions from all coauthors.

### Competing Interests

The authors have no competing interests to declare.

### Acknowledgements

We express our gratitude to the colleagues at Hongshan geomagnetic observatory, Quanzhou geomagnetic observatory, and Yulin geomagnetic observatory for their assistance during the azimuth remeasurement process, which enabled the smooth completion of the survey work.

### Funding Information

Supported by National Natural Science Foundation of China (42374092); The Special Fund of the Institute of Geophysics, China Earthquake Administration (Grant Number DQJB25X26); National Key R&D Program of China (2023YFC3007404).

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
