# Peer review of "The multi scenarios applicability of GNSS differential positioning technology in the remeasurement of observatory azimuth in China"

_EGUsphere, 2025_

## Author Comment (AC3)

**Responds to the comments1**

The authors have properly made corrections in response to my points. Additionally, they have added the discussion with an attempt to correct the vertical deflection, which is welcome. To finalize the paper, please consider the last points below:

L285: Account -> Accounting

L288: 2001). -> 2001),

L290: Where -> where

**Reply:** These errors in L285, L288, and L290 have been corrected.

L302: The paragraph wouldn't start here.

**Reply:** This paragraph has been moved up to L281 in front of Table 3.

Section 5: It is uncommon to have the title "Conclusions and discussion". I suggest that the discussion part (the third paragraph starting from L324) is moved to Section 4 so that Section 5 is dedicated solely for conclusions. Also there is ambiguity in the first sentence of Section 5. Some readers may start with Conclusions, so it is better to clearly describe the "two methods" comparing the azimuths measured in the past and re-measured recently.

**Reply:** Thank you very much for your suggestion. The third paragraph of "Conclusion and Discussion" (starting from L324) discusses the results of the re-measurement. Therefore, it would be more suitable to be placed in Section 4. Now this paragraph has been moved to Section 4 (starting from L281). Section 5 only displays the conclusion section.

**Responds to the comments2**

There is just one comment from Reviewer #2:

It would be better to briefly describe (in Section 4?) how the angle measurements are made with a theodolite (how many repetitions? in both faces?).

**Reply:** Thanks for the reviewer's suggestion. We have added a description of the number of repeated measurements in Section 4 (starting from L288) and provided the mean square error of the measurement results in Scenario II and Scenario IV. These sentences are "The last one is that measurement errors will be introduced when measuring the angles between the azimuth mark and the GNSS target in Scenario II and IV using theodolite. In this work, each angle was measured 6 times. The standard deviation of angle a in Fig. 3 of Scenario II is 2.5", and standard deviation of angle a1 and a2 in Fig. 5b of Scenario IV are 3.3" and 1.8", respectively."

---

## Author Response (AR1)

**Responds to the comments 1**

We sincerely appreciate your valuable suggestions. The author has carefully revised the manuscript based on the feedback, and the modifications have been incorporated into the updated version. Below, we provide detailed explanations regarding the issues raised. The main corrections in the paper and the responds to the comments as flowing are marked in red.

This contribution examines good ways to control the azimuth of a magnetic observatory reference direction, materialized by the reference pillar and a target, usually about 200m distant.

The first method is astronomical, with detailed explanations given about the use of Polaris star. The use of te Sun is mentioned in passing only once.

**Response:** In the text, we have supplemented the description of using the Sun as a celestial body for measurement, referenced the work by Danijel Sugar et al., and enhanced the introduction to astronomical observation methods.

The second method refers to differential GPS, and a special attention is addressed to situations as to how deploy the two GPS receptors on their tripod with respect to the reference pillar and the target.

**Response:** The process of installing two GNSS devices on the straight line formed by the telescope and the azimuth mark is supplemented in the text. The detailed procedure is as follows: First, set up the theodolite on the observation pillar inside the observation room, and precisely adjust it to ensure the instrument's center aligns perfectly with the pillar's center. Then, perform meticulous theodolite leveling. After leveling, aim at the azimuth marker through the theodolite's telescope, and once the crosshairs coincide exactly with the azimuth marker's center, lock the theodolite's horizontal circle at this position (while keeping the vertical circle freely rotatable). Next, mount the leveling base and GNSS target on a tripod, employing a two-person coordination method—one moving the tripod and the other observing through the theodolite's telescope. Position the tripod within the telescope's field of view, then gradually adjust its placement until the GNSS target infinitely approaches the line defined by the crosshairs and the azimuth marker. Secure the tripod afterward. At this point, adjust the leveling base. After leveling, the GNSS target may deviate from the line, requiring another fine adjustment to realign the marker, followed by re-leveling. This cycle must be repeated multiple times until the base is perfectly level and the GNSS target precisely lies on the line. Only after fulfilling these conditions can the GNSS receiver be mounted on this tripod system. The same procedure is applied to the second tripod. Ultimately, ensure the four elements are rigorously collinear.

The authors clearly favor the GPS method (L13) and provide reasons for this:

- Does not require clear sky's and visible celestial body

- Automated calculations

- No timing errors

- No realtime results

However, in my opinion the choice of Polaris in the astronomical method is unfair when doing a competing assessment; the choice of the Sun as the star would bring forward the many advantages of the Sunshot astronomical method especially if a sunshot computer is used, such as the flm5 from RMI (https://www.meteo.be/en/products-and-services/products/flm):

- No need for costly GPS equipment, the observatory DIflux theodolite is sufficient

- No need for two additional GPS stations and tripods installation and alignment

- Automated realtime calculation: the flm5 computes the azimuth of the Sun for any place and time on Earth

- No timing errors with GPS clocks

The real advantage of the GPS is that no celestial body must be visible; it works in cloudy weather. But be aware that the azimuth control is supposed to happen every 10 years, which gives plenty of time for sunshine to materialize.

Then there is the problem of being able to point at the Sun from inside the absolute hut where the reference pillar is rooted. Often windows to the East and West in the hut walls will have been planned at building time so that the Sun is visible at sunrise and/or at sunset, optimal for a sunshot. If not, an external tripod may be installed with view on target and reference pillar.

Considering the precision of both methods, I would point the authors to a seminal paper with detailed comparisons of the methods: "Comparison of the reference mark azimuth determination methods by Danijel Sugar et al., Ann. of Geoph., 55, 6 2012" where the sunshot method is shown to give the required accuracy.

In short, I am unhappy with the insistance of the authors to favor a complicated and costly method (GPS) over a simple and as or more precise one (Sunshot). In the cash strapped environment of magnetic observatories, it should be explained in the text that the same or better accuracy can be obtained using the standard equipment of an observatory.

**Response:** Due to the incomplete description of this work in the paper, there may be some misunderstandings. The original intention of this paper is not to compare which measurement method is superior. Instead, it focuses on the current task to be completed (the re-measurement of azimuth angles at 45 geomagnetic observatories across our country). Through comparison, we aim to illustrate which measurement method—whether in terms of time cost, convenience, or economic cost—is more conducive to completing this task, leading to the selection of the GPS method. Subsequently, we describe several possible scenarios encountered when using this method for measurement. This does not imply that this method is superior to others. We have added more detailed explanations to this section in the text. They are "In the re-measurement work of azimuth markers at geomagnetic observations in Geomagnetic Network of China, we have chosen the GNSS differential positioning technology. The Geomagnetic Network of China comprises 46 geomagnetic observatoties, and most of them have been in operation for over 10 years. The marker's azimuths of the early-constructed observatories were measured using traditional astronomical methods, and usually obtained during the station construction

period. At that time, the observation rooms were not yet roofed, and theodolites could be set up directly on observation pillars to achieve unobstructed visibility of Polaris. Today, however, the rooms have been fully roofed, making it impossible to see the sky from inside. Moreover, the fixed marker pillars outdoors vary in shape (as shown in Fig.3), and are mostly unsuitable for mounting theodolites on them. Furthermore, traditional astronomical observation is highly dependent on weather conditions. Due to the diverse geographical locations of the observatories, especially those in coastal areas, where cloudy weather is frequent It sometimes often takes several days of waiting for suitable weather to conduct astronomical method, making the measurement process time consuming and inconvenient for arranging remeasurement plans. Therefore, after considering factors such as measurement scenarios, time cost, and operational convenience, the GNSS differential positioning method was selected to complete the remeasurement work."

Line indexed comments:

L14: I would remove this sentence as it is not true

**Response:** This statement has been deleted.

L40 and L46: D is the angle between TN and magnetic meridian. Angle between TN an the line connecting...

**Response:** This statement has been modified following this suggestion, and it has been marked in red in the revised draft.

"The magnetic declination (D), defined as the angle between the true north and geomagnetic meridian..."

"the azimuth angle, defined as the angle between the true north and line connecting the center of the observation pillar to the center of the marker..."

L73: this is not true, many new observatories nowadays use astronomy for target azimuth measurement

L86: not true, the Sun is most widely adopted

**Response:** The original statement in the text is inaccurate in expression. The author intended to distinguish between traditional astronomical observation methods, which are based entirely on manual optical theodolites and stopwatches, and modern observation techniques. However, many current observations also rely on more convenient electronic theodolites and GNSS time systems, meaning not all astronomical observations are traditional. Therefore, for clarity, the observation methods in the text are redefined into two categories: traditional and modern. Methods based on electronic theodolites or assisted by modern technology (observing Polaris, the Sun, and other stars) are classified as modern observations. And GNSS technologys are also modern observations.

L93: this is exaggerating the difficulty a lot

**Response:** This sentence has been modified to not make subjective evaluations, but only objectively describe the completed tasks.

L96: Figure 2 is very confusing and possibly wrong. What about S and P?

**Response:** Figure 2 is a schematic diagram of the hour angle method for observing Polaris. To enhance clarity, the revised version substitutes the original partial figure with a complete one, where S denotes Polaris and P represents the North Pole. Corresponding descriptions have also been added in the text. They are "Figure 2 shows the spherical triangle ΔZPS used to calculate the azimuth of Polaris. Based on the altitude angle h of Polaris in the horizontal coordinate system, the latitude $\varphi$ of the observation point, the declination $\delta$ of Polaris, and its hour angle $t$, the formula for calculating the celestial azimuth A (with the positive direction being eastward from north) can be derived using spherical trigonometry:"

L106 and L110: It would be good to have a §discussing how much degradation in accuracy we have

**Response:** The content and data regarding GNSS multipath effects have been supplemented in the text. "they may be affected by the multipath effect (e.g., increased errors near water surfaces, glass curtain walls, and metal reflective surfaces). According to actual observation data, the maximum GPS positioning error caused by the multipath effect can reach 3.4 cm (Fu, 2004). In high-reflection environments, this error can be as large as 15 cm (Wang, 2000). When the distance from a high-reflection surface is approximately 50 meters or more, the multipath effect can basically be neglected (He, 2010)."

L177 and 179: angel -> angle

**Response:** The word has been corrected.

**Responds to the comments 2**

We appreciate the constructive suggestions provided by the reviewers. Based on these feedbacks, we have revised the original manuscript, including modifications to the article title, the rationale for adopting GNSS measurement methods, GNSS measurement errors, the discrepancies between GNSS azimuth angles and those measured by astronomical methods, figure captions, textual expressions, and other aspects. Additionally, we have supplemented relevant content and discussions. The next are the response to the reviewers' comments, and these revisions have been incorporated into the original text, marked in red for clarity.

The paper of He et al., (2025) proposes a GNSS-based method for remeasurement of the azimuth at observatories and presents an overview of the method and some test results. Finally, the paper ranks each method based on qualitative reasoning and concludes that the azimuth has shifted based on the difference between past and current results.

I think that the following points are insufficient.

1. The reason for the authors' decision to use the GNSS method for remeasurement is not clear

GNSS is also needed specialized knowledge such as how to install or software usage (i.e. geodetic expertise). I also wonder if real time is needed. The astronomical method is post-processing, but it does not take much time—around a few ten minutes, in my experience. What is the primary purpose of using GNSS?

**Response:** The authors have supplemented the rationale for adopting the GNSS method in the re-measurement within the text. In the re-measurement work of azimuth markers at geomagnetic observations in Geomagnetic Network of China, we have chosen the GNSS differential positioning technology. The Geomagnetic Network of China comprises 46 geomagnetic observatoties, and most of them have been in operation for over 10 years. The marker's azimuths of the early-constructed observatories were measured using traditional astronomical methods, and usually obtained during the station construction period. At that time, the observation rooms were not yet roofed, and theodolites could be set up directly on observation pillars to achieve unobstructed visibility of Polaris. Today, however, the rooms have been fully roofed, making it impossible to see the sky from inside. Moreover, the fixed marker pillars outdoors vary in shape (as shown in Fig.3), and are mostly unsuitable for mounting theodolites on them. Furthermore, traditional astronomical observation is highly dependent on weather conditions. Due to the diverse geographical locations of the observatories, especially those in coastal areas, where cloudy weather is frequent It sometimes often takes several days of waiting for suitable weather to conduct astronomical method, making the measurement process time consuming and inconvenient for arranging remeasurement plans. Therefore, after considering factors such as measurement scenarios, time cost, and operational convenience, the GNSS differential positioning method was selected to complete the remeasurement work.

2. Systematic errors in azimuth angles obtained using astronomical and geodetic methods.

Are there no errors due to differences in methodology between the azimuth obtained by astronomical methods and the azimuth obtained by GNSS methods? In other words, would the results be the same if the observations of points A and B were replaced with observations of the Polaris, for example? I am concerned about the effect of the Deflection of Vertical (e.g. Vittuari et al., 2016).

**Response:** The reviewed manuscript supplements a discussion on the discrepancies between azimuth angles derived from astronomical methods and those obtained through geodetic methods. Empirical data from practical work is utilized to quantify the errors between these two approaches. Based on this, we further analyze the potential errors in our measurement results.

It should be noted that the azimuth obtained by the astronomical observation method is referenced the plumb line, while the azimuth obtained by the geodetic method (derive from the GNSS method) is based on the ellipsoidal normal. So the vertical deflection introduces systematic error to the azimuths (Vittuari et al., 2016). Account for the influence of vertical deflection, it is necessary to apply the Laplace azimuth equation (9) to convert the astronomical azimuth to the ellipsoidal normal based azimuth (Torge, 2001).

$$a - A = -\eta \tan \varphi + \frac{\eta \cos \alpha - \xi \sin \alpha}{\tan \zeta}$$

Where A is the azimuth determined by astronomic observation and referred to the plumb line, a is the azimuth referred to the ellipsoid normal, $\zeta$ is the ellipsoidal zenith-distance to the reference mark, $\eta$ and $\xi$ are the components of the deflection of the vertical: $\xi = \Phi - \varphi$ and $\eta = (\Lambda - \lambda)\cos\varphi$. They can be calculated by the astronomic coordinates ($\Phi$, $\Lambda$) and the ellipsoidal coordinates ($\varphi$, $\lambda$). Due to the lack of astronomical latitude $\Phi$ and longitude $\Lambda$ values for the measured points, it is temporarily impossible to conduct this calculation. However, previous measurement results can serve as reference for evaluation. For example, Šugar et al. (2012) compared azimuths obtained from astronomical methods and GNSS measurements, finding an average difference of $0.8''$, and the standard deviation of the astronomical azimuth was $\pm 2.6''$. Based on 14 astronomical observations and comparative experiments with GNSS, Wang et al. (2001) reported that the differences between the two methods mostly ranged between $0$–$3''$, with a maximum difference of up to $4.5''$.

I have some question and comment in Section 3.

- Are the distances from the observation pillar same for the two GNSS markers? If not, does this difference affect the results?

**Response:** When two GNSS markers and the azimuth target are aligned in a straight line, the distances from these markers to the observation pillar are unequal. However, by adjusting the telescope focus of the theodolite, the crosshairs can be precisely aligned with the target point. Consequently, the difference in distance between the markers and the observation pillar has minimal impact on the measurement results. The primary source of error stems from the alignment operation of the crosshairs and the target. When the two markers are approximately 200 meters apart, the resulting error is about $1''$. Therefore, the distance between the GNSS markers and observation pillar has negligible effect on the results. As long as the distance between the two GNSS points is sufficiently large and the telescope crosshairs are strictly aligned with their targets, the error can be minimized. Additionally, we have added a detailed procedure for aligning the crosshairs with the target in the original manuscript, which involves iterative adjustments.

- Do GNSS markers obstruct each other when you see them at the observation pillar? If so, how did you arrange the two GNSS markers in a straight line from the observation pillar to the azimuth marker?

**Response:** The observation room floor is typically elevated relative to the outdoor ground. When observing from the observation pillar inside the room towards the outside, the two GNSS markers usually do not obstruct each other. However, if mutual obstruction between GNSS markers occurs, although the two markers can be aligned onto a straight line (i.e., aligning the distant marker first, followed by the near one), this method is not adopted in actual measurements. The reason is that once one marker is obstructed during measurement, real-time verification of alignment becomes impossible. Therefore, when obstruction occurs, Method Scenario I is no longer applicable, and Method Scenario II or alternative measurement scenarios may be chosen instead.

- In the Figure 4 (a) through (d), it is better to explain what each photo shows. That helps for readers to understand this method.

**Response:** For Figure 4, detailed explanatory text has been added in the text."As shown in Fig.4, images illustrate the alignment process from the observation pillar to the GNSS targets and the azimuth marker. Figure 4(a) demonstrates azimuth marker alignment: After precisely centering the azimuth marker using the telescope crosshairs, the horizontal circle should be locked in place. Figure 4(b) shows initial positioning: The tripod with the GNSS target should be aligned with the sightline formed by the telescope crosshairs and the azimuth marker. Figure 4(c) depicts the first aligned GNSS receiver, while Figure 4(d) presents the second aligned GNSS receiver and azimuth marker."

- Are there any restrictions other than distance regarding the location of GNSS markers? For example, are there any obstacles in the vicinity that could cause multipath effect?

**Response:** Yes, in addition to the distance requirements between the two GNSS markers, factors such as multipath effects and potential obstructions cannot be overlooked. These aspects also require attention during measurements, and the author has included additional explanations regarding multipath effects in the article. "They are easy to operate and can work around the clock, but their performance is poor in environments with insufficient signal coverage or obstructions, and they may be affected by the multipath effect (e.g., increased errors near water surfaces, glass curtain walls, and metal reflective surfaces). According to actual observation data, the maximum GPS positioning error caused by the multipath effect can reach 3.4 cm (Fu, 2004). In high-reflection environments, this error can be as large as 15 cm (Wang, 2000). When the distance from a high-reflection surface is approximately 50 meters or more, the multipath effect can basically be neglected (He, 2010)."

I have some question in Section 4
- In the scenario III at Yulin, how short is the distance between the GNSS markers?
- Is the deviation between the original and remeasured values significant compared to the observation error of GNSS? How much error was there in the actual GNSS observations?

**Response:** During the measurement of Scenario IV at Yulin (mislabeled as Scenario III in the original text), the distance between the two GNSS points was approximately 60 m, which may be one of the reasons for the significant error. When using GNSS for measurements, we collected three sets of data and checked the alignment in each measurement. Now, the standard deviation of the three GNSS solutions was added to the result table (Table 2). At the same time, a description of the differences between the astronomical azimuth and the geodetic azimuth measured by GNSS was supplied. And detailed explanations and discussions were given on the final results and the reasons for the large measurement errors in Yulin's scenario IV.

Most of the differences between the re-measured azimuths and the original astronomical azimuths are within the range of 0-9″, that is consistent with theoretical expectations. It should be noted that the azimuth obtained by the astronomical observation method is referenced the plumb line, while the azimuth obtained by the geodetic method (derive from the GNSS method) is based on the ellipsoidal normal. So

the vertical deflection introduces systematic error to the azimuths (Vittuari et al., 2016). Account for the influence of vertical deflection, it is necessary to apply the Laplace azimuth equation to convert the astronomical azimuth to the ellipsoidal normal based azimuth (Torge, 2001). Due to the lack of astronomical latitude and longitude values for the measured points, it is temporarily impossible to conduct this calculation. However, previous measurement results can serve as reference for evaluation. For example, Šugar et al. (2012) compared azimuths obtained from astronomical methods and GNSS measurements, finding an average difference of 0.8″, and the standard deviation of the astronomical azimuth was ±2.6″. Based on 14 astronomical observations and comparative experiments with GNSS, Wang et al. (2001) reported that the differences between the two methods mostly ranged between 0–3″, with a maximum difference of up to 4.5″. Therefore, when considering systematic deviations between astronomical and GNSS methods, positioning errors, sighting alignment errors, and other factors, a discrepancy of 0–9″ is within expected limits. Furthermore, in terms of the accuracy requirement for azimuth in absolute geomagnetic observation (0.1′), a variation of 0–9″ over several decades is considered acceptable. Thus, it can be concluded that no significant displacement has occurred at the observation markers.

However, in the Scenario IV test at Yulin observatory, a significant deviation (>40″) was observed between the measured and original results. Two primary factors may contribute to this result. The first one is the insufficient GNSS baseline length. Despite the RMSE of three measurement sets remained stable at 0.7″, the short distance between two GNSS points (about 60 m) and the maximum horizontal error (about 2.5 mm) may introduce the potential angular error of up to 8.6″. The second one is the theodolite setup error. In this scenario, the theodolite needs to be installed on the tripod at point A, as shown in Fig. 8(b). During angle measurement process, a centering offset was detected on the tripod. After re-centering, a change of 10.8″occurred. This underscores the necessity of adequate baseline distances and highlights the need for further analysis of tripod measurement error characteristics.

The term "observatory" appears at the beginning of the text, but it is unclear whether it refers to observatories in China or observatories around the world. If it refers to the former, it should be clearly stated, and if it refers to the latter, it is necessary to confirm that the information is accurate (e.g. L72).

**Response:** Since the work presented in this paper focuses exclusively on China's geomagnetic observatory network, the "In China" was added to the article title. Thanks for this suggestion.

In citations, there are cases where a space is used before the year and cases where no space is used. This should be standardized.

The following are minor comments with line index:

Lines 17 and 19: The same thing is written on both lines. One of them should be removed.

L20: The "Scenario II" here seems different from the one in Section 3.2. To avoid confusing readers, it is better to use a different name.

L50: Remove the semicolon after "2014."

L69: Punctuation is missing.

**Response:** The author fully accepts the upper suggestions (L17, L19, L20, L50, L69 in the original manuscript) and has made corresponding modifications at the relevant positions.